# Diagnostic Accuracy of Up-Front PET/CT and MRI for Detecting Cervical Lymph Node Metastases in T1–T2 Oral Cavity Cancer—A Prospective Cohort Study

**DOI:** 10.3390/diagnostics13223414

**Published:** 2023-11-09

**Authors:** Christoffer Bing Madsen, Max Rohde, Oke Gerke, Christian Godballe, Jens Ahm Sørensen

**Affiliations:** 1Department of Plastic Surgery, Odense University Hospital, 5000 Odense, Denmark; jens.sorensen@rsyd.dk; 2Research Unit for Plastic Surgery, Department of Clinical Research, University of Southern Denmark, 5230 Odense, Denmark; 3Department of ORL—Head & Neck Surgery and Audiology, Odense University Hospital, 5000 Odense, Denmark; 4Department of Clinical Research, University of Southern Denmark, 5230 Odense, Denmark; 5Department of Nuclear Medicine, Odense University Hospital, 5000 Odense, Denmark; oke.gerke@rsyd.dk; 6Research Unit for Clinical Physiology and Nuclear Medicine, Department of Clinical Research, University of Southern Denmark, 5230 Odense, Denmark

**Keywords:** diagnostic accuracy, MRI, oral cancer, oral squamous cell carcinoma, PET/CT, sensitivity, sentinel node biopsy, specificity

## Abstract

The diagnostic accuracy of up-front 18F-fluorodeoxyglucose positron emission tomography/computed tomography (PET/CT) for detecting cervical lymph node metastases in patients with T1–T2 oral squamous cell carcinoma is reported with large discrepancies across the literature. We investigated the sensitivity, specificity, positive and negative predictive value, and accuracy of up-front PET/CT for detecting cervical lymph node metastases in this patient group and compared the performance to magnetic resonance imaging (MRI). In this prospective cohort study, 76 patients with T1–T2 oral squamous cell carcinoma underwent an up-front PET/CT and MRI at the Odense University Hospital from September 2013 to February 2016. Sentinel node biopsy and elective neck dissection were used for histopathological verification of the imaging modalities. Up-front PET/CT was significantly more sensitive than neck MRI (74% vs. 27%, *p* = 0.0001), but less specific (60% vs. 88%, *p* = 0.001). The accuracy of PET/CT and neck MRI was comparable (66% vs. 63%, *p* = 0.85), the PPV was slightly in favor of neck MRI (56% vs. 62%, *p* = 0.73), the NPV was slightly in favor of PET/CT (77% vs. 63%, *p* = 0.16). Neither PET/CT nor neck MRI should stand alone for N-staging T1–T2 oral cavity cancer.

## 1. Introduction

Cervical lymph node metastases (CLNM) is an important prognostic factor in oral cavity cancer (OCC). The five-year survival rate is reduced by approximately 50% compared to similar primary tumors without neck metastases [1]. Early (T1–T2) oral squamous cell carcinoma (OSCC) has an occult metastatic rate of approximately 30–35% at the cN-staging [2,3,4]. Elective neck dissection (END) has often been used as a gold standard for histopathological verification when investigating the accuracy of a new imaging modality for detecting CLNM [5,6,7,8].

The use of sentinel node biopsy (SNB) is rapidly expanding since multiple international studies have validated SNB as a reliable technique for detecting CLNM for OSCC. Furthermore, the procedure is less invasive compared to END [9,10,11,12]. In addition, several systematic reviews comparing SNB and END in early-stage oral cancer found SNB to be associated with less comorbidities and no significant difference in the comparative effectiveness (overall survival, disease-free survival, and neck recurrence rate) between the two strategies [13,14,15].

Standard imaging modalities for OSCC such as computerized tomography (CT) and magnetic resonance imaging (MRI) have a limited ability to detect occult CLMN [16,17]. Positron emission tomography/computerized tomography (PET/CT) is a relatively new diagnostic tool that is increasingly used in head and neck squamous cell carcinoma (HNSCC). Despite the increasing use of PET/CT in oral cavity cancer, there is a paucity of publications addressing the use of PET/CT scans for detecting cervical lymph node metastases in oral cancer. Studies have reported widespread results and the variations in the study populations and outcomes present a challenge when comparing studies [5,16,18].

The objective of this prospective cohort study was to investigate the sensitivity, specificity, positive and negative predictive value, and accuracy of up-front PET/CT and MRI for detecting cervical lymph node metastases in patients with T1–T2 oral squamous cell carcinoma using sentinel node biopsy or elective neck dissection for histopathological verification.

## 2. Materials and Methods

### 2.1. Study Design

We conducted a prospective cohort study investigating the accuracy of up-front 18F-FDG PET/CT in detecting cervical lymph node metastases in T1–T2 oral squamous cell carcinoma patients with sentinel node biopsy or elective neck dissection for histopathological verification. We included patients with histopathalogically verified T1–T2 OSCC because T3–T4 patients are treated with END per protocol at our institution, due to the high risk of metastasis associated with more progressive stages of malignancy. The diagnostic accuracy of up-front PET/CT was compared to MRI. Our reporting is structured according to the STrengthening the Reporting of OBservational studies in Epidemiology (STROBE) statement [19]. Approval was obtained from the Region of Southern Denmark (journal number 20/9340 and 18/44592) and from the Danish Patient Safety Authority (case number 3-3013-3166/1). The study is based on a subset of patients earlier published by Rohde et al. [20].

### 2.2. Patients

All patients with suspicion of oral cavity cancer referred to the Center of Head and Neck Cancer at Odense University Hospital, Odense, Denmark, from September 2013 to February 2016 were invited to participate in the study. At the initial consultation, the patients received oral and written information regarding the project and signed a consent form if they wished to participate in the study.

In accordance with the local head and neck cancer fast-track program, each patient received the same diagnostic work up that included the following:Examination by an experienced head and neck surgeon, including palpation and ultrasonography of the cervical lymph nodes. Fine needle aspiration was only performed when a strong malignancy suspicion was present.Up-front PET/CT and MRI scans.Biopsy of the primary tumor.Multi-disciplinary team conference.

Inclusion criteria: Patients with histopathologically verified T1–T2 OSCC who underwent a PET/CT, MRI and SNB/END at Odense University Hospital, Odense, Denmark.

Exclusion criteria: Patients who did not undergo, PET/CT, MRI, SNB/END, had a previous oral cancer in the same location, and/or were staged as T3–T4 OSCC. Furthermore, patients with allergy or intolerance to iodine contrast medium, patients in treatment with high doses of systemic steroids (>50 mg/day), patients with reduced kidney function (defined as increased S-creatinine or diagnosed kidney disease) or patients considered unable to cooperate.

### 2.3. Up-Front 18F-FDG PET/CT

As described by Rohde et al. [20], all 18F-FDG PET/CT scans were performed on a hybrid 18F-FDG PET/CT scanner (GE Health Care, Chicago, IL, USA, GE Discovery 690, 710, VCT, or RX). After a 4 h fasting period, 4 MBq/kg of 18F–FDG was administered intravenously. For the PET scan, a standard whole-body acquisition protocol extending from the vertex to the thigh and an acquisition time of 2^1^⁄_2_ min per bed position was used. The head was fixed to prevent movement. PET data were reconstructed into transaxial slices with a matrix size of 128 × 128 (pixel size 5.47 mm) or 256 × 256 (pixel size 2.73 mm), depending on the scanner type. Images from all four PET/CT systems had a slice thickness of 3.27 mm, and reconstruction was carried out by iterative 3D OS-EM.

After the PET scan, multislice diagnostic quality CT scan with intravenous contrast medium (Ultravist^®^ 370 mg/mL, Bayer Healthcare, Leverkusen, Germany) was acquired. The CT scan was obtained with continuous shallow breathing. A standard filter was used to reconstruct data into transaxial slices with a field of view of 50 cm, matrix size of 512 × 512 (pixel size 0.98 mm), and a slice thickness of 3.75 mm. Both PET and CT scans had a 70 cm field of view. A GE Advantage Workstation v. 4.4 or 4.3 or GE AW Server v 3.1 or 3.2 was used for PET/CT analysis.

### 2.4. MRI

MRI was performed on Philips Achieva, Achieva dStream or Ingenia 1.5 T (Philips Medical Systems, Best, The Netherlands) hardware using 20 channel (dStream) head–neck coil as described by Rohde et al. [20]. The exam protocol was kept unchanged for the duration of the study and consisted of STIR, TSE-T2 and -T1 with and without contrast enhancement, in axial or coronal planes with coverage from skull base to aortic arch using 5 mm slices. Diffusion-weighted imaging with spectral fat saturation and apparent diffusion coefficient-maps derived from b-values 0 and 1000 mm^2^/s were carried out on axial 6 mm slices. Images were read on a GE Centricity RA1000 PACS workstation.

### 2.5. Assessment of Scans

The up-front PET/CT scans were evaluated by a team consisting of two experienced radiologists and two nuclear physicians. The MRI scans were evaluated separately by two experienced head and neck radiologists. Characteristics considered in the assessment of lymph nodes were size (enlargement or not), shape (round or not), hilum (fatty/non-fatty), necrosis, center (dense or not), topography of node distribution, and FDG avidity for PET. We did not use a fixed threshold of standard uptake value to determine whether a lesion was malignant or not. The result of the up-front 18F-FDG PET/CT scan was reported as a binary outcome (positive or negative):True positive (TP) was defined as a positive PET/CT and a positive histopathology.True negative (TN) was defined as a negative PET/CT and a negative histopathology.False positive (FP) was defined as a positive PET/CT and a negative histopathology.False negative (FN) was defined as a negative PET/CT and a positive histopathology.

The same classification was applied for the MRI scans. In order for a scan to be classified as true positive, the detected lymph node should be located in the same neck level as the lymph node with a metastsis.

### 2.6. Multi-Disciplinary Team Conference

All patients were discussed at a multi-disciplinary team conference consisting of representatives from plastic surgery (the operating surgeon), oncology, radiology, otorhinolaryngology, and nuclear medicine. An individual treatment plan was designed for each patient based on the results of the biopsy, ultrasonography, MRI, and PET/CT. Treatment plans included surgical removal of the primary tumor, SNB or END and, if needed, surgical reconstruction.

### 2.7. Sentinel Node Biopsy

We used SNB for OCC patients since 2001 at our institution as previously described by Sørensen, et al. [21]. As recommended in the literature, we have used three-dimensional SPECT-CT for high-accuracy identification of the sentinel lymph nodes since 2010 [21,22,23,24]. We used a single-incision surgical approach, followed by frozen sectioning of the SNB, for a perioperative histopathological verification. This setup allows for most of the patients requiring SNB and subsequent END to be treated in a on-estage procedure. The incision from the SNB can be extended in order to provide adequate access for a subsequent neck dissection. Any lymph node with a radioactive count above the background count was removed. We did not use blue dye.

A positive SNB/END is defined as a histopathological verified metastasis, micro-metastasis or isolated tumor cells. Negative SNB/END is defined as histopathologically verified ghost cells or no sign of vital tumor cells. TNM staging was conducted according to the American Joint Committee on Cancer (AJCC) 7th edition.

### 2.8. Statistics

Descriptive statistics were created according to data type: continuous variables were displayed by median (range), categorical variables by frequencies and respective percentages. Two-by-two contingency tables were generated to enable the calculation of sensitivity, specificity, accuracy, positive predictive value (PPV), and negative predictive value (NPV). The McNemar test was performed to assess differences in paired proportions; two-sample proportion test was used otherwise. For sensitivity, specificity, accuracy, PPV, and NPV, exact (Clopper–Pearson-type) 95% confidence intervals were supplemented. *p*-values below 0.05 were considered significant. STATA/MP 17 (StataCorp, College Station, TX, USA) was used for statistical analysis.

## 3. Results

Table 1 describes the characteristics of 76 patients with T1–T2 oral squamous cell carcinoma. Figure 1 provides a detailed overview of the patient inclusion process.

The performance of PET/CT and neck MRI are presented as 2 × 2 contingency tables in Table 2 and Table 3, respectively. The prevalence of cervical lymph node metastases was 41%.

The up-front PET/CT was significantly more sensitive than neck MRI (74% vs. 27%, *p* = 0.0001), but less specific (60% vs. 88%, *p* = 0.001); see Table 4. The accuracy of PET/CT and neck MRI was comparable (66% vs. 63%, *p* = 0.85), the PPV was slightly in favor of neck MRI (56% vs. 62%, *p* = 0.73), and the NPV was slightly in favor of PET/CT (77% vs. 63%, *p* = 0.16).

The MRI found one true-positive CLNM that the PET/CT did not find. The MRI found 14 true-negative CLNM that the PET/CT did not find; however the PET/CT found one true-negative CLNM that the MRI did not find. Eight patients had CLNM despite two negative scans. Four patients had disease-free necks despite two positive scans.

A positive SNB was found in 16 patients, seven were found intraoperatively on frozen sections and the remaining nine were found postoperatively on paraffin sections.

One patient received adjuvant radiotherapy instead of END due to personal preference. A total of 15 patients received a completion node dissection due to a positive SNB. It was not possible to identify and harvest a sentinel node in one patient (<2%); therefore, the procedure was converted to an END. Six out of twenty-one patients (29%) with suspicious lymph nodes, who received END, had no lymph node metastasis.

## 4. Discussion

### 4.1. Main Findings

To our knowledge, this is the largest prospective cohort study to date that exclusively investigated the accuracy of up-front 18F-FDG PET/CT in T1–T2 oral squamous cell carcinoma. This study demonstrated improved sensitivity by up-front PET/CT for the detection of cervical lymph node metastases compared to MRI (74% vs. 27%, *p* = 0.0001). On the contrary, up-front PET/CT was inferior to neck MRI in terms of specificity (88% vs. 60%, *p* = 0.001).

### 4.2. Strengths and Limitations in Relation to Other Studies

Linz et al. [25] recently published a prospective study where they found the sensitivity, specificity, PPV, and NPV for PET/CT in the detection of CLNM in 135 T1–T4 OCC patients to be 82.4%, 83.5%, 65.1%, and 92.7% respectively. They found the MRI to have a sensitivity of 70.6%, specificity of 62.6%, PPV of 41.4%, and NPV of 85.1%. When comparing the two imaging modalities, the only parameter that did not result in a significant difference was the sensitivity. Linz et al. found the PET/CT to be significantly more specific than the MRI, which is in contrast with our findings. Both the study by Linz, et al. [25] and our study reported a similar sensitivity (82.4% vs. 74%) and NPV (85.1% vs. 77%). The largest discrepancy in the PET/CT performance was the specificity (83.5% vs. 60%). The MRI performance was not in accordance with our results. The largest difference was found in sensitivity (70.6% vs. 27%). A reason for the abovementioned discrepancies could be their larger study population and, especially, their inclusion of T3–T4 tumors.

Niu et al. [26] investigated the accuracy of PET/CT in detection of neck metastases of OSCC in patients without large palpable lymph nodes. They reported a sensitivity, specificity, accuracy, PPV, and NPV of PET/CT of 84%, 73%, 77%, 59%, and 91%, respectively. Their results were based on 78 T1–T4 OSCC patients. Compared with our results, they showed a higher sensitivity, specificity, accuracy, and NPV, whereas their PPV was similar to our result. They included the use of maximum standardized uptake value (SUV_max_) in their study and concluded that it was useful in diagnosing cervical lymph node metastases. The difference between our results may well be due to their use of SUV_max_ combined with their inclusion of patients with T3–T4 OSCC (32% of their cohort), as we know that more advanced tumors will produce larger CLNMs that are easier to detect [27].

Unfortunately, we did not use SUV when analyzing our PET/CT scans. However, we believe that further investigations should measure different SUV parameters to help standardize diagnostic algorithms in order to correctly identify neck disease in the future.

Bae et al. [16] investigated the performance of PET/CT for detecting CLNM in 178 patients with T1–T4 OCC. They found the PET/CT to have a sensitivity of 69.1%, specificity of 77.9%, accuracy of 75.8%, PPV of 49.2%, and NPV of 89.1%. We evaluated PET/CT as being more sensitive (74%), but less specific (27%) compared to their results. Again, their inclusion of T3–T4 tumors is likely to contribute to the difference seen in our results. Their higher specificity could be explained by the fact that larger tumors produce larger CLNMs, which, in turn, are easier to detect [27].

Thoenissen et al. [28] conducted a retrospective study where 137 T1–T4 OCC patients received a preoperative neck MRI for N-staging. They reported a sensitivity of 66% and a specificity of 68%. These results clearly differ from our respective findings (27% and 88%).

Laimer et al. [29] and Souren et al. [30] report a high MRI sensitivity when detecting CLNM in OCC (85.7% and 83.1%, respectively). Moreover, they also report considerably high specificities (75.6% and 75.7%). As mentioned before, the inclusion of T3–T4 tumors is likely to contribute to the difference seen compared to our results, but does not explain our higher specificity of 88%.

Table 5 summarizes some recent reports of sensitivity and specificity for PET/CT and/or MRI in detecting cervical lymph node metastases, based on an ad hoc literature search in MEDLINE/PubMed. The search terms were “(Sensitivity and specificity) AND (PET/CT) AND (lymph node metastases) AND (MRI) AND (oral cancer OR oral cavity cancer OR oral squamous cell carcinoma)”.

### 4.3. Strengths and Limitations

To our knowledge, this is the largest prospective cohort study exclusively investigating the accuracy of up-front PET/CT in T1–T2 oral squamous cell carcinoma. Both the PET/CT and MRI scans were performed on the same day, and the assessments were conducted by experts masked to the results of the other imaging modalities. We consecutively included patients over a two-and-a-half-year period, and each patient acted as his/her/their own control when comparing the imaging modalities. The strengths and validity of paired diagnostic studies versus randomized (controlled) studies are well-described in the literature [31,32,33,34]. By using a paired design, we reduce the risk of confounding. Additionally, a paired design offers advantages such as dependence of the sample size on the agreement rate between the modalities, the multiple aims of diagnostic accuracy studies, and the possibility of the early unmasking of results at the individual level. Finally, our study set-up was identical to the clinical practice and, consequently, the results may be more generalizable and externally valid [35].

A limitation of our study is our inconsistency in treatment: 21 patients who presented with T1–T2 OSCC were treated with an END instead of a SNB. One patient was converted from SNB to END because it was not possible to detect a sentinel lymph node. In nine patients, the deviation from SNB was due to suspicion of cervical lymph node metastases on both ultrasonography and PET/CT. One patient received END based on a positive ultrasound. Finally, 10 patient treatment plans were changed from SNB to END because of a positive PET/CT.

All patients were discussed at a multi-disciplinary team conference involving a panel of experts in head and neck oncology who agreed on the individual treatment plans. Unfortunately, there was an inconsistency in MDT decisions where, in some instances, a positive PET/CT alone could result in an END.

In more recent years, depth of invasion (DOI) has been implemented as an important factor in oral cancer. DOI was implemented in the 8th edition of the AJCC Cancer Staging Manual by including DOI in the T-staging of oral cavity cancer. Furthermore, The National Comprehensive Cancer Network (NCCN) recommends elective neck dissection in cases with a DOI greater than 4 mm [36,37]. Our data collection took place from 2013 to 2016 and, at that time, DOI was not commonly used at our center and was unfortunately not incorporated into our study as a standard measure.

Despite the abovementioned recommendation to perform END for DOI > 4 mm, a randomized multicenter non-inferiority trial from Japan [38] compared sentinel lymph node biopsy-navigated neck dissection with elective neck dissection for T1–T2, node-negative, no distant metastasis, with ≥4 mm (T1) depth of invasion. The patients were randomly assigned to undergo END (*n* = 137) or sentinel lymph node biopsy-navigated neck dissection (*n* = 134). Their primary endpoint was the 3-year overall survival rate. Secondary endpoints included postoperative neck functionality and complications and 3-year disease-free survival. Sentinel lymph nodes underwent intraoperative multislice frozen section analysis for diagnosis. Patients with positive sentinel lymph nodes underwent either one-stage or second-look neck dissection. The study concluded that SNB-navigated neck dissection may replace END without a survival disadvantage and reduce postoperative neck disability in patients with early-stage OCSCC.

The use of DOI would have been an interesting aspect in our study, despite the abovementioned evidence that SNB-navigated neck dissection is non-inferior to END, even with DOI > 4 mm. Future studies investigating the accuracy of up-front PET/CT and MRI in detecting clinically occult neck disease in T1–T2 oral cavity cancer should include DOI in order to meet the newest international guidelines.

### 4.4. Challenges

There are a couple of challenges in using PET/CT for staging the cervical lymph nodes. The first challenge is the high rate of false-positive results. In our study, 18/76 (23%) patients had a false-positive result, which can potentially lead to a more invasive treatment. Other studies have reported similarly high false-positive results. LeRose, et al. [39] reported 15/84 (18%) false-positive PET/CT results in their study population. Their estimated proportion of false-positive results was marginally smaller than ours, which could be explained by the inclusion of T1-T4N1-N3 head and neck cancer. Inflammation is known to increase the metabolism of the tissue, resulting in a higher accumulation of 18F-FDG and, consequently, a hotspot on the PET/CT scan, which can be mistaken for an increased metabolism due to malignancy. When designing individual treatment plans for patients, it is important to acknowledge the relatively high number of false positives in order to correctly assess the impact of a positive PET/CT scan.

The second challenge is the limited ability of PET/CT to detect micrometastases (≤2 mm in largest diameter) for the detection of CLNM in T1–T2 OSCC. Back in the year 2000, Crippa et al. [40] found that FDG PET can detect lymph node metastases >10 mm in diameter in 100% of cases. However, the detection rate fell to 83% in metastases size 6–10 mm and 23% for metastases size < 5 mm. Even though the quality of PET/CT scans and analysis tools has improved since then, micrometastases are still responsible for many of the false-negative test results in cervical lymph node staging in head and neck squamous cell carcinoma. During our data collection period (2013–2016), the pixel size improved from 128 × 128 pixels to 256 × 256 pixels. However, the greatest advancements have been the hardware in the scanners used to fuse and create the PET/CT images. Even with a continuation of these improvements, PET/CT is seemingly still not good enough at detecting micrometastases in the cervical lymph nodes. We reported a false-negative result in 8/76 (10%) patients. This is slightly lower than the findings in other studies. LeRose et al. [39] reported 14/84 (17%) false negatives. An explanation for this limitation may be that PET/CT largely depends on the number of cells that can metabolize glucose, whereas SNB is independent of the number of cells in a micrometastases and is capable of detecting even isolated tumor cells. A recent systematic review and metanalysis on the rate of occult neck metastasis in anterior tongue cancers highlights the importance of studies reporting the findings on early stage anterior tongue cancers (almost half of our study population) as their high rate of occult neck metastesis remains a problem in N-staging [41].

The third challenge is that our MRI sensitivity seems inferior to other studies in detecting cervical lymph node metastases. This influences our ability to make recommendations based on our findings. We lack a suitable explanation for our low MRI sensitivity. To our knowledge, we are the only study that investigated T1–T2 OCC. All other studies to which we compared our MRI performance included T1–T4 OCC. We believe that the larger primary tumors produce larger CLNMs that are easier to detect on a MRI scan. Finally, 10 patients had CLNM despite a negative PET/CT and a negative MRI. This is the primary challenge when N-staging early-stage oral cancer. The high rate of occult CLNM is still a major problem. A recent systemtic review and meta-analysis of the occult neck metastasis rate of T1–T2 antiror tongue cancer, which occurred in 45% of our study population, argues that more studies should describe their experience with this group, since the correct N-staging of this group remains a challenge despite the recent developments in imaging modlities. This could help to establish more precise knowledge of the accuracy of our diagnostic imaging modlities for N-staging oral cancer, which is necsessary to create an international consensus on the optimal staging and treatment of this patient group [41].

### 4.5. Perspectives

PET/CT has been repeatedly proven to be efficient at finding patients with disease; however, PET/CT scanning may come with a high number of false positives. MRI is precise at finding patients without disease, but may lead to a high number of false negatives. In other terms, a negative PET/CT (relatively high sensitivity) rules out disease, whereas a positive MRI (high specificity) rules in disease [42]. In addition to N-staging, the use of PET/CT provides information on the dissemination of the disease.

In a prospective cohort study from 2017, Rohde et al. [43] concluded that PET/CT has a significantly higher detection rate of distant metastases or synchronous cancers than chest CT/head and neck MRI for patients with oral cavity cancer (T1–T4). The detection of distant metastasis and synchronous cancers at the time of diagnosis has a significant impact on treatment decisions and prognosis [20,44]. This qualifies more for a valid argument in favor of up-front PET/CT than of an improved sensitivity in detecting cervical lymph node metastases, as PET/CT has the significant impact of M-classification, thus affecting treatment planning for patients with OSCC.

## 5. Conclusions

Up-front PET/CT was significantly more sensitive, but less specific, than neck MRI in the detection of cervical lymph node metastases of patients with T1–T2cN0 oral cavity cancer. Neither PET/CT nor MRI should stand alone for the N-staging of patients with T1–T2 oral squamous cell carcinoma.

## Figures and Tables

**Figure 1 diagnostics-13-03414-f001:**
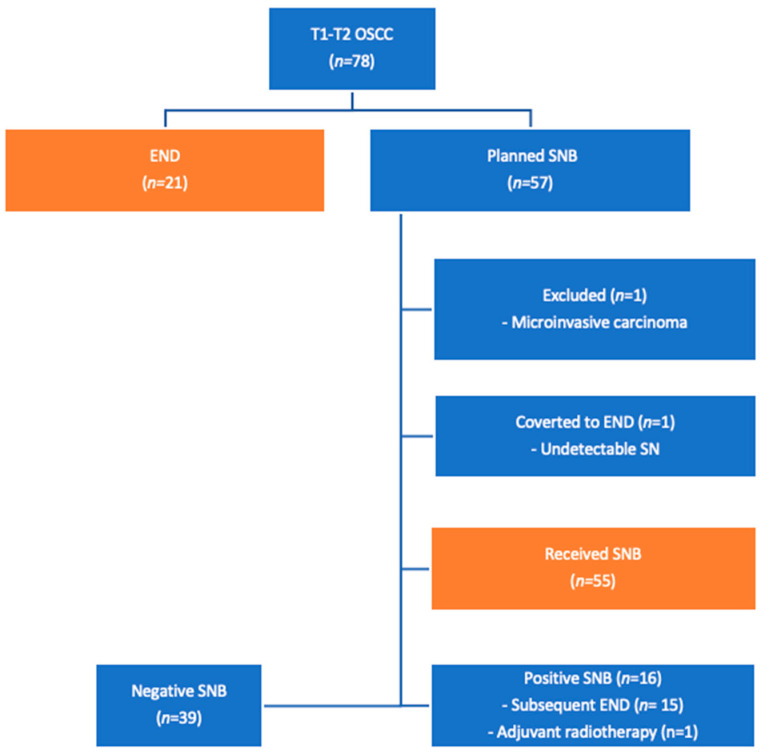
Inclusion, exclusion, and treatment of patients with oral cavity cancer. The orange boxes represent the final study population. The blue boxes provide additional information on the process to help understand the orange boxes and their results. Twenty-one patients received END and fifty-five patients initially underwent SNB. The multi-disciplinary team conference agreed that the preoperative imaging strongly indicated neck disease of a larger magnitude than just the sentinel lymph node in the 21 patients who received END without a forgoing SNB.

**Table 1 diagnostics-13-03414-t001:** Patient characteristics.

Variable	Descriptive Statistics
Number of patients	76
Sex	Male: 52 (68%)
	Female: 24 (32%)
Median age (range)	63 years (21–85 years)
Location of primary tumor	Floor of mouth: 27 (36%)
	2/3 anterior tongue: 34 (45%)
	Upper alveolus: 1 (1%)
	Lower alveolus: 7 (9%)
	Buccal mucosa: 5 (7%)
	Hard palate: 1 (1%)
	Retromolar area: 1 (1%)
T-stage	T1: 53 (70%)
	T2: 23 (30%)
Median tumor size (range)	18 mm (5–40 mm)
Regional * recurrence rate	SNB: 9/55 (16%)
	END: 8/21 (38%)
Median follow up (range)	SNB: 57 months (1–70 months)
	END: 30 months (3–80 months)

* We have defined regional as an oral or cervical recurrence. No distant recurrence was observed.

**Table 2 diagnostics-13-03414-t002:** 2 × 2 contingency table for PET/CT (*n* = 76).

PET/CT	Histopathology	
	Positive	Negative
Positive	23 TP	18 FP
Negative	8 FN	27 TN

**Table 3 diagnostics-13-03414-t003:** 2 × 2 contingency table for neck MRI (*n* = 73) s ^1^.

Neck MRI	Histopathology	
	Positive	Negative
Positive	8 TP	5 FP
Negative	22 FN	38 TN

^1^ Three patients were not scanned due to claustrophobia.

**Table 4 diagnostics-13-03414-t004:** Comparison of PET/CT and neck MRI in 76 T1–T2cN0 oral squamous cell carcinoma patients. All results regard cervical lymph node metastasis alone.

Parameter	PET/CT	Neck MRI	*p*-Value
Sensitivity (95% CI)	74% (55–88%)	27% (12–46%)	0.0001
Specificity (95% CI)	60% (44–74%)	88% (75–96%)	0.001
Accuracy (95% CI)	66% (54–76%)	63% (51–74%)	0.85
PPV (95% CI)	56% (40–72%)	62% (32–86%)	0.73
NPV (95% CI)	77% (60–90%)	63% (50–75%)	0.16

**Table 5 diagnostics-13-03414-t005:** Overview of the most recent publications and their results on the sensitivity and specificity of PET/CT and MRI for detecting cervical lymph node metastases in oral cavity cancer.

Publication	Population	Year	PET		Neck MRI	
			Sensitivity	Specificity	Sensitivity	Specificity
Bae et al. [16]	T1–T4 OCC	2020	69.1%	77.9%		
Niu et al. [26]	T1–T4 OCC	2020	83.9%	73.1%		
Thoenissen et al. [28]	T1–T4 OCC	2023			66%	68%
Laimer et al. [29]	T1–T4 OCC	2020			85.7%	75.6%
Souren et al. [30]	T1–T4 OCC	2015			83.1%	75.7%
Linz et al. [25]	T1–T4 OCC	2021	82.4%	83.5%	70.6%	62.6%
Our results	T1–T2 OCC		74%	60%	27%	88%

## Data Availability

The data are unavailable due to legal restrictions.

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
