# Peer review of "Diagnostic Accuracy of Up-Front PET/CT and MRI for Detecting Cervical Lymph Node Metastases in T1–T2 Oral Cavity Cancer—A Prospective Cohort Study"

_diagnostics, 2023, doi:10.3390/diagnostics13223414_

Round 1

Reviewer 1 Report

Comments and Suggestions for Authors

1.     In the present study, authors enrolled 76 patients with T1-T2 oral squamous cell carcinoma. Is T1-T2 clinical stage or pathological stage? Please clarify

2.     The criteria of positive lymph node in MRI and PET/CT should be described in detail in the method section for clinician readers.

3.     Authors described in the method section that “True positive (TP) was defined as a positive PET/CT and a positive histopathology. True negative (TN) was defined as a negative PET/CT and a negative histopathology. False positive (FP) was defined as a positive PET/CT and a negative histopathology. False negative (FN) was defined as a negative PET/CT and a positive histopathology.” If the positive lymph node location by histopathology is different from positive lymph node location by PET/CT, does it belong to TP, TN, FP, or FN? For example, if the positive lymph node location by histopathology is level III and positive lymph node location by PET/CT is level II, does it belong to TP, TN, FP, or FN? Please clarify.

4.     Table 1 shows 2/3 anterior tongue: 34 (45%). Do these 34 patients have p16 immunohistochemistry? How about their HPV status (p16 immunohistochemistry status)?

5.     In table 1, SNB: 9/55, END: 8/21. There was total 76 patients. But, in figure 1, END=21, SNB=57. There was total 78 patients. Are there 76 or 78 patients? Please clarify.

6.     Eight patients (about 10%) had CLNM despite two negative scans. It seems high. Do they have T1 or T2 disease? Do authors suggest routine SNB or END in these patients? Please discuss

7.     Do patients with tongue cancer have higher rate of lymph node metastases than others?

8.     In table 5, authors list other series report, but they areT1-4 OCC. Did these papers have subgroup T1-2 OCC data? IF yes, I think it is better to list their T1-2 OCC data as comparison.

9.     Extranodal extension (extracapsular spread) is a very poor prognosticator. How about the extranodal extension in the present study? How about the value of PET/CT or MRI in detection of extranodal extension?

Comments on the Quality of English Language

nil

Author Response

  1. In the present study, authors enrolled 76 patients with T1-T2 oral squamous cell carcinoma. Is T1-T2 clinical stage or pathological stage? Please clarify.

Response: Thank you for your comment. As seen in table 1, 78 patients were initially included due to their clinical T1-T2 staging. Seventy-six patients were included in the final study population due to their pathological T1-T2 staging. For example, one patient was excluded due to microinvasive carcinoma. We have clarified in our this in our methods section.

  1. The criteria of positive lymph node in MRI and PET/CT should be described in detail in the method section for clinician readers.

Response: Thank you for bringing this to our attention. We have clarified how the scans were assessed in our method section under the subheading: “Assessment of Scans”.
For MRI the lymph nodes were evaluated based on size (enlargement or not), shape (round or not), hilum (fatty/non-fatty), necrosis, center (dense or not), topography of node distribution.
For PET/CT scans the FDG avidity was also used. As described, we did not use a fixed threshold of standard uptake value to determine whether a lesion was malignant or not. The limitations of this is also discussed in our discussion section.

  1. Authors described in the method section that “True positive (TP) was defined as a positive PET/CT and a positive histopathology. True negative (TN) was defined as a negative PET/CT and a negative histopathology. False positive (FP) was defined as a positive PET/CT and a negative histopathology. False negative (FN) was defined as a negative PET/CT and a positive histopathology.” If the positive lymph node location by histopathology is different from positive lymph node location by PET/CT, does it belong to TP, TN, FP, or FN? For example, if the positive lymph node location by histopathology is level III and positive lymph node location by PET/CT is level II, does it belong to TP, TN, FP, or FN? Please clarify.

      Response: Thank you for your comment. If a lymph node in level II was positive on the PET/CT and the histopathology from level II was without any sign of metastasis, then we would classify the PET/CT as false positive.  If an END was performed and the histopathology from level III is with a lymph node metastasis and the PET/CT did not show any sign of disease in level III, then the PET/CT would be classified as a false negative. We did not experience cases similar to your example so our answer is hypothetical. We have clarified in our methods section under the subheading “assessments of scans”.

  1. Table 1 shows 2/3 anterior tongue: 34 (45%). Do these 34 patients have p16 immunohistochemistry? How about their HPV status (p16 immunohistochemistry status)?

Response: Thank you very much for your comment. No, unfortunately we do not have sufficient data on p16 immunohistochemistry status to include the data in this article. If we had the data, we would present it as we agree it would strengthen the article.

  1. In table 1, SNB: 9/55, END: 8/21. There was total 76 patients. But, in figure 1, END=21, SNB=57. There was total 78 patients. Are there 76 or 78 patients? Please clarify.

Response: Thank you for brining this to our attention. We apologize for the unclarity. The final study population included 76 patients. We have highlighted the boxes with the final study population with orange and clarified this in the text “Inclusion, exclusion, and treatment of patients with oral cavity cancer. The orange boxes represent the final study population. The 21 patients who received END and 55 patients who initially underwent SNB.”

  1. Eight patients (about 10%) had CLNM despite two negative scans. It seems high. Do they have T1 or T2 disease? Do authors suggest routine SNB or END in these patients? Please discuss.

Response: thank you very much for you comment. We agree that this is high – too high. That is one of the reasons that we need studies like this, to evaluate the imaging modalities used for N-staging in early-stage oral cancer. In our introduction we state that standard imaging modalities for OSCC such as computerized tomography (CT) and magnetic resonance imaging (MRI) have a limited capability of detecting occult CLMN. PET/CT is increasingly used for TNM staging in oral cancer, but limited evidence is available in the litterateur and the existing literature present widespread results on sensitivity, specificity, and predictive values for PET/CT to detect CLNM. In our discussion we mention the main problem with the PET/CT being the large amount of false positive. The eight patients in our study were both T1 and T2 but they represent the main challenge with staging earl oral cancer. Thirty to thirty-five percent have occult CLNM. The significant decrease in the 5-year survival rate when CLNM are present is the reason for the importance for correct N staging of the neck.
We have added a few lines on this in our discussion.

  1. Do patients with tongue cancer have higher rate of lymph node metastases than others?

Response: thank you for your comment. To our knowledge anterior tongue cancers have a higher rate of occult lymph node metastases compared to other oral cavity cancers. Furthermore, they are known for their irregular pattern of metastasis. Our institution has previously described the lymphatic patterns of oral cancer metastases and anterior tongue cancers were the only primary tumor with sentinel lymph nodes in neck level IV. We have added a section on this in our discussion with relevant references. 

  1. In table 5, authors list other series report, but they areT1-4 OCC. Did these papers have subgroup T1-2 OCC data? IF yes, I think it is better to list their T1-2 OCC data as comparison.

Response: thank you for your comment. Yes, several studies have subgroups including T1-T2 oral cancer, however, they only report the pooled values for sensitivity, specificity, and predictive values on the total study population. If they had presented relevant outcomes for T1-T2 would present these data as we agree this would be relevant.

  1. Extranodal extension (extracapsular spread) is a very poor prognosticator. How about the extranodal extension in the present study? How about the value of PET/CT or MRI in detection of extranodal extension?

Response: Thank you for your comment. We have not done an analysis of the sensitivity, specificity, and predictive value for the subgroup of patients with extracapsular spread. This could be an interesting analysis to include in future studies.

Reviewer 2 Report

Comments and Suggestions for Authors

Abstract:

1.       Clear and Concise Language: The abstract should be more concise and to the point. Remove unnecessary phrases like "Citation: To be added by editorial staff during production," and publisher notes. The abstract should summarize the study findings and its importance.

2.       Include Key Findings: Mention the key findings of the study in the abstract. This could include the main results related to sensitivity, specificity, and predictive values of PET/CT and MRI.

3.       Highlight Clinical Significance: Provide a sentence or two in the abstract about the clinical significance of the study. For instance, mention how the findings can impact patient care or clinical practice.

4.       Methods Mention: Include a brief mention of the methods used in the study, such as the number of patients, the verification process, and the period of the study.

Introduction:

5.       Clear Background Information: The introduction could benefit from a more concise presentation of background information. For instance, briefly explain the importance of detecting cervical lymph node metastases in T1-T2 oral cavity cancer and its impact on survival.

6.       Cite Relevant Studies: When discussing the background, cite relevant studies or statistics that support the information you present. For example, you mention a reduction in the five-year survival rate - you should provide a reference for this statistic.

7.       Use Subheadings: Consider using subheadings to structure the introduction, making it easier for readers to follow your line of thought. For example, you can have sections like "Importance of CLNM in OCC," "Verification Methods," and "Limitations of Existing Imaging Modalities."

8.       Provide a Clear Research Objective: Clearly state the research objective of your study in the introduction. What specific information are you seeking to provide or verify with this research?

9.       Include a Hypothesis or Research Questions: If applicable, include the research questions or hypothesis that guided your study.

10.   Explain the Gap in Existing Research: Discuss why your study is necessary by pointing out gaps or limitations in existing research. For example, why is the accuracy of PET/CT for detecting cervical lymph node metastases in T1-T2 oral cavity cancer an important area of investigation?

11.   Conclude with Study Rationale: Conclude the introduction with a statement explaining the rationale for your study, why it is needed, and what you hope to contribute to the existing knowledge in this area.

Methods

12.   Study Design:

·         Consider providing a brief rationale for choosing T1-T2 patients and explain how this choice aligns with the study's objectives and clinical relevance.

·         Mention the exact number of patients included in the study.

13.   Patients:

·         Consider providing more information about the demographics of the patients, such as age, gender, and any other relevant characteristics.

·         Clarify whether these patients were newly diagnosed cases or if they had already undergone treatment.

14.   Up-front 18F-FDG PET/CT:

·         Clarify the specific criteria for patient fasting before the PET scan, as this can affect the accuracy of the results.

·         Include information on the PET/CT scanner used (e.g., manufacturer, model) for transparency.

2.4 MRI:

·         Specify the MRI protocol and settings in more detail, including the sequences used, magnetic field strength, and any specific parameters (e.g., TE, TR, FOV).

·         Mention any contrast agents used during MRI, including the type and dosage.

15.   Assessment of Scans:

·         Provide more information about the qualifications and experience of the radiologists and nuclear physicians who evaluated the PET/CT scans.

·         Explain why a fixed threshold of standard uptake value was not used, and what criteria were employed to determine malignancy based on FDG avidity.

16.   Multi-disciplinary Team Conference:

·         Mention the frequency and format of these conferences (e.g., weekly meetings).

·         Explain the decision-making process at these conferences and how the results of the diagnostic tests were integrated into treatment planning.

17.   Sentinel Node Biopsy

·         Provide a brief description of the single-incision surgical approach and the process of frozen sectioning.

·         Clarify the method for radioactive count measurement and the criteria for deciding whether a lymph node should be removed.

18.   Statistics

·         Specify which variables were considered continuous and which were categorical.

·         Mention if there were any potential sources of bias or limitations in the statistical analysis.

Comments on the Quality of English Language

Moderate changes required

Author Response

Abstract

  1. Clear and Concise Language: The abstract should be more concise and to the point. Remove unnecessary phrases like "Citation: To be added by editorial staff during production," and publisher notes. The abstract should summarize the study findings and its importance.’’

Response: Thank you for your suggestions. The abovementioned phrasing is a part of the template used for submission in Diagnostics. We do not feel comfortable editing the template.

  1. Include Key Findings: Mention the key findings of the study in the abstract. This could include the main results related to sensitivity, specificity, and predictive values of PET/CT and MRI.

Response: Thank you very much for your feedback. We have made sure that the sensitivity, specificity and predictive values for MRI and PET/CT are presented in the abstract.

  1. Highlight Clinical Significance: Provide a sentence or two in the abstract about the clinical significance of the study. For instance, mention how the findings can impact patient care or clinical practice.

Response: We appreciate this suggestion, thank you very much. We have added a few lines on the clinical significance at the end of our abstract. Neither PET/CT nor neck MRI should stand alone for N-staging T1-T2 oral cavity cancer.”

  1. Methods Mention: Include a brief mention of the methods used in the study, such as the number of patients, the verification process, and the period of the study.

Response: Thank you for this comment. We have made sure that the exact study period is specified in the methods section under the subheading “patients”. The number of patients is presented in our results section and in table 1.

Introduction

  1. Clear Background Information: The introduction could benefit from a more concise presentation of background information. For instance, briefly explain the importance of detecting cervical lymph node metastases in T1-T2 oral cavity cancer and its impact on survival.

Response: Thank you for your feedback. We have done our best to be more concise in our presentation of the background information and highlighting the impact of cervical lymph nodes metastases in T1-T2 oral cancer.

  1. Cite Relevant Studies: When discussing the background, cite relevant studies or statistics that support the information you present. For example, you mention a reduction in the five-year survival rate - you should provide a reference for this statistic.

Response: Thank you very much for your comment. We have made sure that there is a sufficient number of citations to support the information presented in the introduction. Furthermore, we have made sure that the statistic from your example is followed up with a reference.

  1. Use Subheadings: Consider using subheadings to structure the introduction, making it easier for readers to follow your line of thought. For example, you can have sections like "Importance of CLNM in OCC," "Verification Methods," and "Limitations of Existing Imaging Modalities."

Response: We appreciate the suggestion. We have tried to clarify our line of thought and better structure our introduction without using subheadings but by emphasizing the key point in each paragraph.

  1. Provide a Clear Research Objective: Clearly state the research objective of your study in the introduction. What specific information are you seeking to provide or verify with this research?

Response: Thank you very much for your feedback. We have done our best to clarify the research objective. We have explicitly stated the objective at the end of the introduction.

  1. Include a Hypothesis or Research Questions: If applicable, include the research questions or hypothesis that guided your study.

Response: Thank you for your comment. We have adjusted accordingly.

Hypothesis: MRI and PET/CT both have limited accuracy. MRI is limited by its high number of false negative outcomes and PET/CT by its high number of false positive outcomes.

  1. Explain the Gap in Existing Research: Discuss why your study is necessary by pointing out gaps or limitations in existing research. For example, why is the accuracy of PET/CT for detecting cervical lymph node metastases in T1-T2 oral cavity cancer an important area of investigation?

Response: Thank you for your comment. We believe that the existing literature tends to overestimate the accuracy of PET/CT due to their reporting of pooled results including T1-T4 oral cancers. T1-T2 oral cavity cancer are known to have a rate of occult CLNM around 30-35 %. END would result in overtreatment of the remaining 70-65%. However, the 5-year survival-rate is reduced with 50% if CLNM is present. Thus, making correct N-staging of the group very important. We have clarified in the article.

  1. Conclude with Study Rationale: Conclude the introduction with a statement explaining the rationale for your study, why it is needed, and what you hope to contribute to the existing knowledge in this area.

Thank you for your comment, we have clarified our rationale for the study.

Methods

  1. Study Design:
  • Consider providing a brief rationale for choosing T1-T2 patients and explain how this choice aligns with the study's objectives and clinical relevance.

Response: Thank you for your comment. Due to the increased risk of CLNM with T3-T4 tumors these patients are treated with END per protocol at our institution. T1-T2 oral cavity cancer are known to have a rate of occult CLNM around 30-35 %. END would result in overtreatment of the remaining 70-65%. However, the 5-year survival-rate is reduced with 50% if CLNM is present. Thus, making correct N-staging of the group very important.

  • Mention the exact number of patients included in the study.

Response: thank you for your comment. The study population consisted of 76 patients. We have reported this in the results section, Table 1, and Figure 1.

  1. Patients:
  • Consider providing more information about the demographics of the patients, such as age, gender, and any other relevant characteristics.

Response: Thank you for your comment, please see table 1 for relevant patient characteristics.

  • Clarify whether these patients were newly diagnosed cases or if they had already undergone treatment.

Thank you for your comment, the patients are newly diagnosed and presenting with their first oral cancer. We have clarified this in our methods section under the subheading “patients”.

  1. Up-front 18F-FDG PET/CT:
  • Clarify the specific criteria for patient fasting before the PET scan, as this can affect the accuracy of the results.

Response: se comments below.

  • Include information on the PET/CT scanner used (e.g., manufacturer, model) for transparency.

Thank you for your comment. We have described these details in the methods section under the subheading “Up-front 18F-FDG PET/CT”. scans were performed on a hybrid 18F-FDG PET/CT scanner (GE Discovery 690, 710, VCT, or RX). After a 4-hour 104 fasting period, 4 MBq/kg of 18F–FDG was administered intravenously. For the PET scan, a standard whole-body acquisition protocol extending from the vertex to the thigh and an acquisition time of 21⁄2 min per bed position was used. The head was fixed to prevent movement. PET data were reconstructed into transaxial slices with a matrix size of 128 × 128 (pixel size 5.47 mm) or 256 × 256 (pixel size 2.73 mm), depending on the scanner type. Images from all four PET/CT systems had a slice thickness of 3.27 mm, and reconstruction was done by iterative 3D OS-EM. After the PET scan, multislice diagnostic quality CT scan with intravenous contrast medium (Ultravist® 370 mg/mL) was acquired. The CT scan was obtained with continuous shallow breathing. A standard filter was used to reconstruct data into transaxial slices with a field of view of 50 cm, matrix size of 512 × 512 (pixel size 0.98 mm), and a slice thickness of 3.75 mm. Both PET and CT scans had a 70 cm field of view. A GE Advantage Workstation v. 4.4 or 4.3 or GE AW Server v 3.1 or 3.2 was used for PET/CT analysis.

2.4 MRI:

  • Specify the MRI protocol and settings in more detail, including the sequences used, magnetic field strength, and any specific parameters (e.g., TE, TR, FOV).

Response: thank you for your comment. We have described the details that we noted from the data inclusion period. Unfortunately, we don’t have more information than already provided in the methods section under the subheading “MRI”.

  • Mention any contrast agents used during MRI, including the type and dosage.

Response: thank you for your comment. As described in the article’s method section under the subheading “MRI” the exam protocol was kept unchanged for the duration of the study and consisted of STIR, TSE-T2 and -T1 with and without contrast enhancement, in axial or coronal planes with coverage from skull base to aortic arch using 5 mm slices. We have not reported type and dosage, unfortunately.

  1. Assessment of Scans:
  • Provide more information about the qualifications and experience of the radiologists and nuclear physicians who evaluated the PET/CT scans.

Response: thank you for your comment. As described in the article: “The up-front PET/CT scans were evaluated by a team consisting of two experienced radiologists and two nuclear physicians.” The whole team where at senior consultant level, thus giving them many years of experience in their given field and their head and neck cancer subspeciality.

  • Explain why a fixed threshold of standard uptake value was not used, and what criteria were employed to determine malignancy based on FDG avidity.

As described in out methods section under the subheading “Study design” the study is based on a subset of patients earlier published by Rohde, et al. To our knowledge the reason for not using a fixed threshold of standard uptake value because there was a lack of international consensus at the time of data collection on what an optimal fixed threshold of standard uptake value should be in order to detect CLNM in oral cavity cancer.

  1. Multi-disciplinary Team Conference:

Mention the frequency and format of these conferences (e.g., weekly meetings).

Response: Thank you for your comment. The meetings are held 3 times a week: Monday, Wednesday, and Friday. The format is as described in the methods section under the subheading “Multi-disciplinary Team Conference”: “Representatives from plastic surgery (the operating surgeon), oncology, radiology, otorhinolaryngology, and nuclear medicine discuss each patient. All the relevant information regarding each patient such as pathology, imaging and the performance status are presented and are included in the final treatment plan.”

  • Explain the decision-making process at these conferences and how the results of the diagnostic tests were integrated into treatment planning.

Response: Thank you for your comment. All patients are evaluated in the multi-disciplinary team meetings. Most patients are treated per protocol. For example, a histopathological verified oral squamous cell carcinoma with a cN0 neck (negative PET/CT, negative MRI, negative UL, no palpable cervical lymphadenopathy) will be treated with a resection of the tumor and SNB. Sometimes the patient’s results are more divergent. Perhaps the UL and MRI are without suspicion of CLNM, but the PET/CT shows FGD avidity on the neck. Then the team will discuss pros and cons and together make a decision based on the available data. The decision process is individualized to fit each patient with the whole patient in mind including performance status etc. thus making it difficult to describe the exact decision-making process at these conferences.

  1. Sentinel Node Biopsy
  • Provide a brief description of the single-incision surgical approach and the process of frozen sectioning.

Response: thank you for your comment. The single-incision surgical approach is previously described by our institution, but as suggested we have provided a more detailed description in our methods section under the subheading “sentinel node biopsy”.

Regarding the process of frozen sectioning, we initiate the surgical procedure by harvesting the sentinel node biopsy. The specimen is sent to the Department of Pathology where it is analyzed while we remove the oral cancer. We have elaborated as recommended.

Clarify the method for radioactive count measurement and the criteria for deciding whether a lymph node should be removed.

Response: Thank you for your comment. As mentioned in our methods section under the subheading “sentinel node biopsy”: “Any lymph node with a radioactive count above the background count is removed.”

  1. Statistics

Specify which variables were considered continuous and which were categorical.

Response: thank you for your comment. Categorical variables were data such as sex (male/female), location of primary tumor (FOM/ant. tongue/upper alveolus), and T-stage (T1/T2). Continuous variables were data such as median age, median tumor size, and median follow-up. We have described how the two different types of variables are presented in the methods section under the subheading “statistics”: “Descriptive statistics were done according to data type: continuous variables were displayed by median (range), categorical variables by frequencies and respective percentages.

Mention if there were any potential sources of bias or limitations in the statistical analysis.

Response: Thank you for your feedback. The statistical analysis used in our study are considered lege artis. As mentioned in our discussion “The strengths and validity of paired diagnostic studies versus randomized (controlled) studies are well described in the literature [31-34]. By using a paired design, we reduce the risk of confounding. Additionally, a paired design offers advantages such as dependence of the sample size on the agreement rate between the modalities, the multiple aims of diagnostic accuracy studies, and the possibility of early unmasking of results at the individual level. Finally, our study set-up was identical to the clinical practice and, consequently, the results may be more generalizable and externally valid [35]. “

Round 2

Reviewer 1 Report

Comments and Suggestions for Authors

Aors have answered my questions. I have no further questions.